# Designing a library of lived experience for mental health (LoLEM): protocol for integrating a realist synthesis and experience based codesign approach

Fiona Lobban,[1] Paul Marshall,[1] John Barbrook,[2] Grace Collins,[3] Sheena Foster,[1] Zoe Glossop ,[1] Clare Inkster,[4] Paul Jebb,[5] Rose Johnston,[1] Hameed Khan,[1] Christopher Lodge,[1] Karen Machin,[6] Erin Michalak,[7] Sarah Powell ,[8] Jo Rycroft-Malone,[9] Mike Slade,[10,11] Lesley Whittaker,[5] Steven H Jones[1]

For numbered affiliations see end of article.

**Correspondence to**
Dr Fiona Lobban;
f.lobban@lancaster.ac.uk

## ABSTRACT

**Introduction** People with lived expertise in managing mental health challenges can be an important source of knowledge and support for other people facing similar challenges, and for carers to learn how best to help. However, opportunities for sharing lived expertise are limited. Living libraries support people with lived expertise to be 'living books', sharing their experiences in dialogue with 'readers' who can ask questions. Living libraries have been piloted worldwide in health-related contexts but without a clear model of how they work or rigorous evaluation of their impacts. We aim to develop a programme theory about how a living library could be used to improve mental health outcomes, using this theory to codesign an implementation guide that can be evaluated across different contexts.

**Methods and analysis** We will use a novel integration of realist synthesis and experience-based codesign (EBCD) to produce a programme theory about how living libraries work and a theory and experience informed guide to establishing a library of lived experience for mental health (LoLEM). Two workstreams will run concurrently: (1) a realist synthesis of literature on living libraries, combined with stakeholder interviews, will produce several programme theories; theories will be developed collaboratively with an expert advisory group of stakeholders who have hosted or taken part in a living library and will form our initial analysis framework; a systematic search will identify literature about living libraries; data will be coded into our analysis framework, and we will use retroductive reasoning to explain living libraries' impacts across multiple contexts. Individual stakeholder interviews will help refine and test theories; (2) data from workstream 1 will inform 10 EBCD workshops with people with experience of managing mental health difficulties and health professionals to produce a LoLEM implementation guide; data from this process will also inform the theory in workstream 1.

**Ethics and dissemination** Ethical approval was granted by Coventry and Warwick National Health Service Research Ethics Committee on 29 December 2021 (reference number 305975). The programme theory and implementation guide will be published as open access

### STRENGTHS AND LIMITATIONS OF THIS STUDY

⇒ Consistent with the Medical Research Council guidance for development of complex interventions, this protocol describes a novel integration of a realist synthesis to develop a programme theory, with a series of experience-based codesign workshops to develop an implementation guide for a living library.

⇒ The study design is creative, iterative and involves relevant stakeholders, including people with lived expertise, in all aspects of design and delivery.

⇒ The theory about how the living library will work to improve outcomes and the utility of the implementation guide both need to be tested and refined in future research in which the library of lived experience for mental health is delivered across a range of contexts.

and shared widely through a knowledge exchange event, a study website, mental health provider and peer support networks, peer reviewed journals and a funders report.

**PROSPERO registration details** CRD42022312789.

## INTRODUCTION

Approximately one in six adults and children in the UK experience mental health difficulties at any one time,[1 2] with similar rates reported around the world.[3] Despite drug treatments and psychological interventions on offer, there is no evidence of these rates reducing. Indeed, following the impacts of the COVID-19 pandemic, rates are likely to increase.[4] Health services are already unable to meet demand. New and inclusive ways to help society manage mental health are needed.

The value of learning to manage health from lived expertise is well recognised globally, pioneered by voluntary and third-sector organisations. Value comes from the knowledge and support from people who have

walked the same path but also from the radically different relationship between the person sharing their expertise and those seeking help, which is one of mutual empowerment and growth, rather than professional-to-patient treatment.[5]

Currently, there are several opportunities for people to share their lived expertise to help others. Around the world, committees addressing policy development, research funding and clinical guidelines (eg, National Institute for Health and Care Excellence guidelines in the UK) include people with lived experience. Training courses for mental healthcare professionals have lived experience reference groups contributing to curricula, and in mental health services, people are employed at a strategic level as peer leaders, members of patient/carer councils, and increasingly in the UK as peer support workers (PSWs), supported by growing evidence (eg, see Gillard et al[6]), and a competency framework and national training programme.[7] However, evidence of the impacts of lived expertise support on patient outcomes is still limited,[8 9] and mechanisms of change are poorly understood. Further, all of these roles require significant investments of time, restricting them to people able to make this their job. This excludes the vast majority of people and reduces the diversity of experience being shared.

More research is needed to develop and evaluate models of involvement that can facilitate wider participation of those from diverse backgrounds (including people in work/with caring responsibilities). Here we examine one example: the living library model.

In a living library (also referred to as a human library), 'readers' are people invited to select living 'books' from a catalogue of synopses (provided by the books) and have a conversation. During this, they are encouraged to ask questions about things that matter to them. Hence, the library is not just about telling stories but also about dialogue to inform, challenge our preconceptions and change the way we think. Books, readers and librarians have equal status, have the right to decline to answer any question and may end the conversation if they wish. The living library concept is well established in the Human Library,[10] originally face-to-face in Copenhagen, but now also online. However, living libraries have been used around the world across a range of health and non-health-related contexts[11] but, to our knowledge, are yet to be embedded in any sustained mental health support services.

A living library for mental health, offered as a sustained service, and shown to be effective, could overcome many limitations of existing involvement frameworks. People from diverse backgrounds or with seldom heard voices could train to become a book, and flex their availability around other commitments, especially if the library was online and available beyond traditional working hours. Dialogue within the library enables readers from a range of backgrounds to explore directly the issues that they care about, and revisit conversations with books where appropriate. It allows the books to shape their stories and to share what they feel the reader needs to understand, which may be more effective than reading static written stories. Books can be trained and supported to create and share their stories in ways that feel safe and empowering for both books and readers. They remain in control of what is shared with who, how, and when, and have direct experience of the immediate impacts of their sharing. Interacting face-to-face (or via video link) may reduce our tendency to see people who have experienced mental health difficulties as different from us and as 'others', facilitating empathy and reducing stigma.[12] Finally, the library metaphor may help engagement of books and readers, as libraries are familiar places, with a focus on enhancing well-being, e.g.,[13] frequently attended,[14] and positively perceived[15] by people experiencing mental distress.

The idea of a living library to improve mental health outcomes has international appeal and has already been tried around the world in a range of contexts, including improving self-management strategies for people diagnosed with bipolar disorder,[16] reducing stigma in pharmacy students in Canada[17]; and improving mental health literacy in students in Hong Kong.[18] Living books have reported a wide range of benefits including; a greater self-awareness,[19] personal enjoyment and enhanced social relationships,[20 21] catharsis from sharing experiences[22] and feeling valued in promoting change.[23] For readers, there is evidence for enhanced self-efficacy through discovering new ways to understand and manage specific difficulties.[16] Direct contact for carers (formal and informal) with people with lived experience in an equal status context, can increase empathy, knowledge, perceived similarity, and reduce stigma and anxiety about contact.[24]

However, living libraries also have the potential for negative impacts. All libraries are hosted within an organisational system, which in turns sits within a broader social context. Such contexts are never neutral, and can determine which stories are shared, how, and with what impacts.[25] As has been shown with the use of recovery narratives in mental health services, even when the opportunity to share lived expertise stories is intended to be empowering and to promote diversity, the use of recovery stories in health and social contexts can serve to propagate normative organisational agendas through the selection and honing of stories to allow only those that are 'acceptable'.[26] Storytellers may fear unwanted consequences of sharing their experiences, such as loss of personal freedom, especially if they have previously been treated under a section of the mental health act.[27] The process of becoming a living book may be stressful, triggering traumatic experiences, or may confirm, rather than challenge existing negative stereotypes if living libraries are not well designed and books are not given equal status to readers and library staff.[28]

Our aim is to explore how a living library could be set up and used to train health and social care professionals, support people experiencing mental health difficulties

and their carers, and reduce stigma in the general population across the UK. To do this, first we need to understand the likely impacts, both positive and negative, for books and readers, how these come about, for whom and in what context. Building this programme theory of how a living library is likely to work is a key early step in the Medical Research Council framework for the development of complex interventions[29] and ensures that we design the library within contexts that will successfully trigger key mechanisms of positive change while reducing any potential harm. It also guides us to select ways of assessing important impacts (positive and negative) for both readers and books. Second, the library needs to be codesigned by a team of relevant stakeholders including people with lived expertise, mental health service providers, librarians and commissioners. The processes of building the theory and codesigning the library protocol must run in parallel so each can usefully inform the other. The evolving programme theory can identify contextual factors that need to be addressed in the codesign of the library. Reciprocally, issues raised by stakeholders during the codesign process can identify contextual factors that need to be theorised to understand how they will have impact on the success (or otherwise) of the library.

To meet our objectives, we will use a novel integration of a realist synthesis to build our programme theory and a modified version of the experience-based codesign (EBCD) approach to develop the library protocol.

Realist synthesis is a theory-driven methodology that aims to explain how and why observed outcomes occur in different contexts.[30] The approach is grounded in realist philosophy and proposes that complex interventions in health have intended and unintended impacts (outcomes (O)) through the ways in which people respond to the programme resources offered (mechanisms (M)). The extent to which these mechanisms are triggered depends on specific features of the environment in which the programme is delivered or characteristics of the participants (contexts (C)). Realist synthesis is a useful early step in developing theory-informed interventions in healthcare. However, the processes by which the theories are developed and then used to inform the design process are often not well articulated. In this study, we show how a realist synthesis can be run in parallel with a modified version of the EBCD approach to develop interventions which are both theory-informed and fully grounded in lived experience.

EBCD is a qualitative approach which captures in-depth experiences of stakeholders, identifies emotionally charged points in the care process and uses these as a focus for codesigning changes to care to improve outcomes.[31] Positive impacts of the process have been reported by people taking part in the process within a mental health context (eg, see Springham and Robert[32]). EBCD has largely been conducted with the intention of service improvement but has been recognised as having potential for early-stage intervention design projects, when combined with programme theory development.[33]

For example, Fylan *et al* provide a detailed description of how they integrated EBCD with the use of behaviour change theories and methods to develop and test interventions for safer medicine use across healthcare settings.[34] However, we were unable to identify another study that combined EBCD with a realist synthesis process. Therefore, we have outlined our protocol in detail and followed Preferred Reporting Items for Systematic Review and Meta-Analysis Protocols guidelines,[35] in the hope this will be of use to others in developing other theory-informed interventions.

### Aims and objectives
Our aim is to codesign a theory-informed library of lived experience for mental health (LoLEM)

Our objectives are
► To build a programme theory for the LoLEM using a realist synthesis of research and practice literature with primary data from interviews with people who have been involved in living libraries. Our theory will include likely positive and negative impacts of the LoLEM on books and readers across a range of different contexts.
► To codesign a guide for delivering a LoLEM. Through a series of codesign workshops with a multistakeholder group, and drawing on the developing theory, we will design key features of the LoLEM, outlining core components needed to successfully deliver and evaluate a living library and highlighting flexible adaptation to aid implementation and effectiveness across different contexts.

Our research questions are
► What are the impacts of a LoLEM on books and readers? How do these impacts happen, for whom and in what contexts?
► What are the key aspects to consider in designing a LoLEM?

Our team includes people with experience in group facilitation and peer support for mental health (KM and CL), mental health professionals (PJ and LW), clinical academics (FL, SJ and JR-M), information specialists (JB), researchers (PM, RJ and ZG) and an artist (GC). We have established a non-independent expert group to oversee and guide the project. This includes five people with expertise in hosting (EM, CI and SP), researching (MS) or taking part in a living library as a book with experience in managing mental health challenges (HK), and a mental health carer with a broad interest in lived expertise frameworks (SF). The group will be consulted at each of the steps described further throughout the study, via monthly online video meetings.

### METHODS AND ANALYSIS
Our study is under way and will run from the end of September 2021 to the end of January 2023. We will run two workstreams in parallel to ensure each informs the other. Weekly operational meetings for each stream will

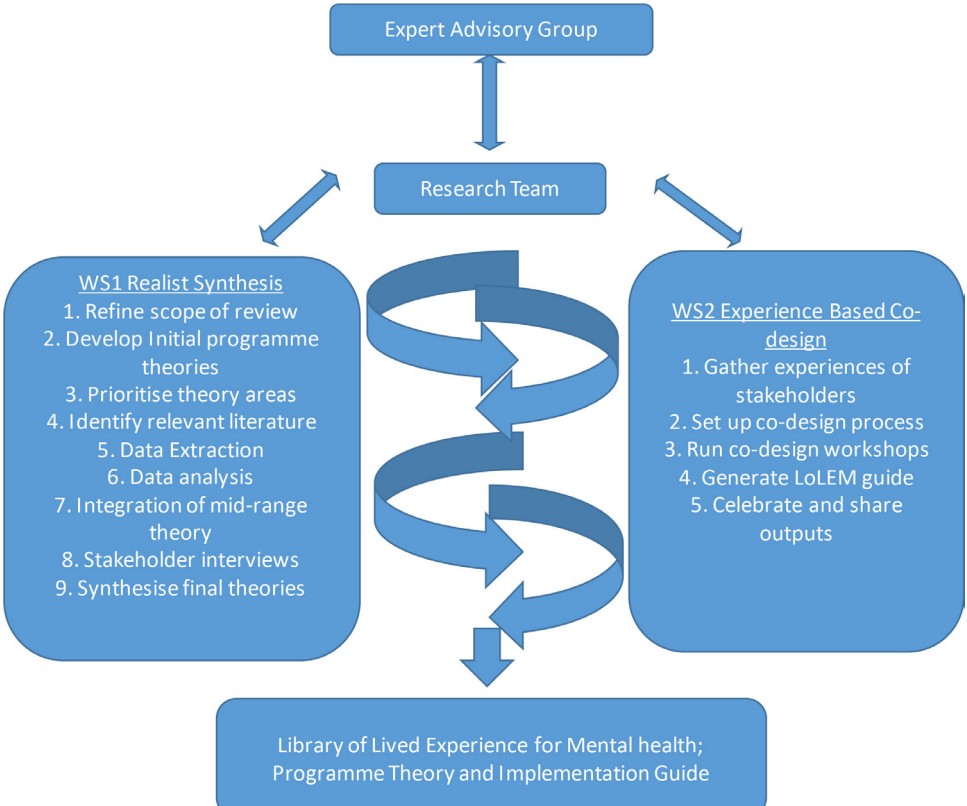

**Figure 1** Library of lived experience for mental health study design.

guide activity and facilitate synergy. Monthly meetings with the expert group will support strategic direction and problem solving, and ensure this work is situated within the broader context of peer support in mental health. The study design is shown in figure 1.

### Workstream 1: realist synthesis

The aim of our realist synthesis is to develop a programme theory about what the likely impacts would be of a LoLEM, for who, how and in what context. Following established realist methodology,[30 36] we will develop our programme theory of how living libraries 'work', drawing on relevant 'middle-range' theories, reviewing relevant literature and conducting in-depth interviews with key stakeholders. Key steps are described in detail as follows.

#### Refining the scope of the review

The scope of our review was refined through early discussion with our expert group at the outset of the study. The aims were to estimate the size of relevant literature, specify terms to include in our search strategy, interview stakeholders and identify key papers for our search sensitivity analysis.

We undertook a broad scoping search of relevant electronic databases including Web of Science, PsycINFO, MEDLINE, SocINDEX and CINAHL, using search terms "living librar*" OR "human librar*" OR "live librar*" OR "living book*" OR "human book*" OR "people book*" OR "live book*". We searched grey literature through Google/Google scholar (top 100 results), OpenGrey (multidisciplinary) and ProQuest (theses and dissertations) using search terms "living library" OR "human library". We screened references and citations for key papers. We identified 44 published articles that reported an evaluation of a living library: 10 from a healthcare care context (9 journals and 1 report) and the rest in non-health (educational) contexts. The majority of studies were small-scale qualitative evaluations of the impacts of temporary living libraries. There were many more reports of living libraries being run but not evaluated. Findings showed that the living library concept has high face validity, but more rigorous research is needed to test impacts and underlying mechanisms in different contexts.

Through discussion with our expert group, the scope of the review was set as in table 1.

Given the limited data available, we did not limit the scope to living libraries within a mental health context.

#### Developing initial programme theories

We will draw on lived experiences of the research team and expert group to develop our initial programme theories. We will first conduct individual interviews with members who have hosted a living library (CI), or have been a human book (CL and HK) or a reader (FL and SJ), and refine emerging theories in group discussion.

We will generate initial propositional statements in the form of 'if… then' statements to begin to identify causal links between concepts,[37] full context–mechanism–outcome (CMO) statements, or 'theory areas',

| Table 1 | Scope of the review |
|---|---|
| Intervention | For the purpose of this review, living libraries (also referred to as human libraries) are defined as<br>▶ Synchronous interaction between a human 'book' and 'reader'.<br>▶ Conversations are time-limited and supported/hosted by 'librarians'.<br>▶ Conversations relate to any aspect of psychosocial functioning/well-being.<br>▶ Held under the name 'living library' or 'human library'. |
| Context | Any living library in any setting. We will focus on any features of the library, backdrop against which it is being delivered or the individual members that may influence how the books/readers/librarians respond to the library. This includes but is not limited to individual characteristics of people involved, organisational setup and location of the library, how books and readers are recruited, how books are trained and supported, and target population for readers. |
| Mechanism | The response of the books and readers to the resources offered by the library |
| Outcome | Any positive or negative impact on books or readers or librarians |

that is, broad concepts related to context, mechanism or outcomes, that are considered important.

### Prioritisation of theory areas

Based on previous experience of this methodology, we anticipate generating a high number of initial programme theories and/or theory areas. With the expert group, we will prioritise theories around which to build our analysis framework using NVivo V.12.[38] The prioritisation process will continue throughout the study. At this stage, we anticipate prioritising theories that (1) are specific to the living library context, rather than theories which would be applicable to any peer-based intervention, (2) attempt to explain both positive and negative impacts of the library, and (3) address pragmatic issues around implementation identified in EBCD workshops.

### Identifying relevant literature

Working with our library information specialist (JB), we will develop an initial search strategy that can be adapted for use across several electronic databases and can be refined iteratively as our theories develop.

Our first search will identify literature that can inform our theory about how the LoLEM might work. Given the limited literature available, we will include living libraries in any context not specific to mental health. There are likely to be similarities (but also important differences) in underlying mechanisms of how living libraries work across contexts.

We will select multidisciplinary research databases in addition to relevant health, social science and education databases to maximise search coverage, including Web of Science, Scopus, PsycINFO, MEDLINE, CINAHL, Embase, SocINDEX, Social Care Online, British Education Index, Education Abstracts, Library, and Information Science & Technology Abstracts.

Databases will be searched using the following terms: "living librar*" OR "human librar*", applied in each database to 'title', 'abstract', and where available 'keywords' fields. Grey literature will be searched using: ProQuest (theses/conference documents), British Library via Ethos (theses), International Clinical Trials Registry Platform (clinical trials), Overton.io (UK health policy), Bielefeld Academic Search Engine (articles/these/conference documents), Google.com*.

We will conduct google searches of the following domains for websites that included the words "human library" or "living library". - Inurl: ac (academic institutions) - Inurl: edu (academic institutions) - Inurl: gov (government-published) - Inurl: org (non-governmental organisations) - Inurl: nhs (UK National Health Service).

We will test the sensitivity of the search strategy by ensuring it picks up five key papers identified from our literature scoping in each of the electronic databases that includes the paper. We will include a wide range of data sources including primary data, reviews, commentaries, manuals and detailed lived experience accounts. Inclusion and exclusion criteria are listed in box 1.

All identified documents will be uploaded to EndNote and duplicates will be removed. Deduplicated records will be uploaded into Rayyan[39] to facilitate a two-step screening process.

In step 1, all documents will be screened by title and abstract with 20% screened by a second reviewer to check reliability. Any uncertainties will be discussed and refinements will be made to criterion descriptors as needed.

At stage 2, full texts will be read, first to check the inclusion criteria are met, and second to assess the additional criteria of rigour and relevance. Rigour will be assessed by the overall quality of the study or the theoretical arguments being put forward, and the confidence we have in the conclusions being drawn. Relevance will be assessed by the extent to which the document tells us anything useful to refine, refute or elaborate our programme

> **Box 1 Inclusion criteria for living library systematic search screening**
>
> **Inclusion criteria**
> ⇒ Full text available in English.
> ⇒ Published since 2000 (inception of the human library concept).
> ⇒ Relates to human or living library (as per the aforementioned definition).
> ⇒ Any study type is permitted.
> ⇒ Written text only (excluding audiovisual content).

theories.[40] Only studies considered to be relevant and of sufficient rigour for us to trust the implications for our theory development will be included. We will not use formal assessment of methodological quality tools often used in systematic reviews. The value of individual studies to the development of the programme theory in a realist synthesis depends not only on the methodological rigour but also on what the document can tell us that is relevant to building our theory, and the conceptual thinking of the authors.

We will conduct hand searches of reference sections (backward searching) and citations (forward searching) of eligible documents included in the review after this full-text screening stage. We will also consult members of our expert advisory group for any additional materials that may have been missed by our search strategy.

## Data extraction
Using a proforma designed for the study, we will extract details about each included study including publication details (authors, date and title); study details (location, design and sample characteristics); and intervention details (where applicable, format of the intervention). Reading the full paper in detail and extracting data from all sections, we will extract any data that pertains to any of our identified theory areas and code these extracts into our NVivo framework. We do not anticipate identifying full explicit CMO configurations but will use inductive, deductive and abductive reasoning to work back from what is being said and infer what the underlying CMO configurations may be and code the data accordingly.[41] Where data relevant to our theory are identified but are not easily coded into our theory areas, we will refine and add to our framework. At this point, data from both work-streams will be integrated. Field notes and online inter-active whiteboards created during the EBCD workshops will also be coded into the NVivo framework and will continue to inform the theory development throughout the rest of the study.

## Data analysis
A detailed analysis of each theory area will be done using retroductive reasoning. This involves working back from the data to identify the context-dependent mechanisms underlying the impacts described in the data that can help us to understand the impacts of living libraries across a range of different contexts and why these impacts are occurring. Analysis will focus on refining, reconfiguring, elaborating, refuting and consolidating each of the theory areas described in the NVivo framework. Here we will draw on several strategies including

► Comparing instances in which a living library has been used in very different contexts to understand how and why the outcomes are different.
► Examining extreme cases where very positive or very negative outcomes are reported for a library or any specific individuals within the library.

► Examining unintended and unexpected outcomes to determine what contextual factors may have triggered mechanisms to generate these.
► Using counterfactual thinking and thought experiments to consider whether a particular outcome could have occurred without a specific contextual factor being present, what it is about this contextual factor that leads to the outcome and imagining how the outcome might have been different if a specific aspect of context was modified.

We hope this process will refine our initial programme theories, but we anticipate it will also generate further questions.

## Drawing on formal theories
To further elaborate our theories and fill in any gaps in our CMO configurations, we will draw on formal theories that can explain how a LoLEM might work by theorising relevant mechanisms at a level of abstraction above the specifics of the living library model. For example, at the individual (micro) and interpersonal (meso) levels, these may include intergroup contact theory,[42 43] social identity theory,[44] social learning theory[45] and personal recovery.[46] Additional theories at the organisational (macro) level may also be relevant, such as the theory of health empowerment.[47] At this stage, and consistent with the iterative nature of realist synthesis methodology, we may conduct additional theory-driven literature searches for further formal theories that augment the theory development process. The selection and use of formal theories will be guided by the extent to which each theory helps us to further develop our programme theories.

## Stakeholder interviews
Our second strategy to elaborate and refine our theories, will be to use realist interviewing[48] with key stakeholders. We will sample through snowballing techniques, initially drawing on the networks of our broad research team, expert group and key individuals identified through our literature searching. Early interviews will inform the development of our initial programme theories, but as our theories develop, the focus of the interviews will evolve into a more specific focus on filling in gaps in our theories and testing competing theories. Sampling will become more purposive to select interviewees with relevant knowledge and expertise. Those willing to take part will be sent a participant information sheet and asked to provide consent online. Interviews will be done flexibly (to maximise participation) using individual/group interviews, online/face-to-face/telephone media, and topic guides designed to test our explanatory theories. Data will be recorded and transcribed, and coded and analysed within the theory framework in NVivo as described in steps 5 and 6.

## Synthesising final theories
In the final stage of the synthesis, we will use iterative discussion within our research team and expert group to

consolidate our programme theories into a final framework that articulates how a LoLEM works, for whom, and in what contexts. This will draw on data derived from literature searches, realist interviews, and the EBCD workshops described further.

## Workstream 2: EBCD of a LoLEM guide

Informed by the EBCD approach,[49] we will adapt the process to meet the specific needs of this project. Key steps are outlined as follows.

### Gathering experiences from stakeholders to understand key issues to be addressed

In EBCD, these are referred to as 'touchpoints' and are identified through interviews, observations or group discussion. In this project, we will use the data from steps 1–3 of the realist synthesis to identify key features of living libraries that need to be codesigned.

### Setting up the codesign process

Participants will be recruited through NHS Trusts, peer support charities and existing user involvement networks across the North of England to take part in a series of monthly codesign workshops. Participants will include people with experience of using their mental health experiences to support others or to inform research and development in health services and mental health professionals with expertise in supporting peer workers. All participants will be sent a participant information sheet and will be asked to consent to data from the workshops being used to inform study outputs. Participants will be paid in accordance with guidance from the National Institute of Health Research[50] and will be offered access to individual support from the research team. Workshops will be run online to maximise engagement geographically and through pandemic restrictions. They will be cofacilitated by an experienced trainer in peer support interventions whose work is grounded in lived expertise (KM), our PPI lead for the project (CL) and an artist (GC) with experience in codelivering and capturing codesign processes using creative visual tools. The process will be supported by other members of the research team.

### Running the codesign workshops

The content of each workshop will be responsive to our developing programme theories and agreed in consultation with participants during the process. The initial plan consists of four phases to the process:

► Building rapport and trust within the group through experiences of story sharing which are fundamental to the living library model.
► Applying the experiences of story sharing to generate further understanding of the living library model and to identify barriers and facilitators to success.
► Applying the understanding of the barriers and facilitators to develop fundamental guidelines and key considerations for running a living library in a range of settings.

► Agreeing on the format of production of the guidelines and next steps.

### Generating a LoLEM guide for how to set up and run a LoLEM across a range of contexts

The aim is not to write a 'recipe book' but to identify key things to consider when setting up a library and to understand how different decisions are likely to lead to different outcomes. These will be described using text and artistic impressions. Key issues anticipated include what the library is for, where and how the library is hosted, how books are defined, recruited, trained, reimbursed, and supported; who the readers are, and how the library is offered to them; how the role of the librarian is set up; and how a LoLEM should be evaluated.

### Celebrating and sharing outputs

A celebration event will be held to share the final draft of the protocol and to review the process of codesign.

## Patient and public involvement (PPI) statement

Lived expertise is embedded at every level of the project. During the study design, and prior to applying for funding, we held an online consultation with four service users from our Spectrum Centre Advisory Panel at Lancaster University,[51] which informed the idea of a living library (two had been living books). Perceived benefits were that this approach can be engaging, non-stigmatising and draws on a familiar and non-threatening metaphor. It has the potential to involve a wide range of books and readers due to flexibility in time/location/ what is shared/ways of sharing, and the opportunity for in-depth dialogue. Challenges identified were the need for adequate training, supervision and support for books, as well as ensuring empowerment, ownership and control by people with lived experience. While designing the study, we hosted an online stakeholder event (n=8) with services users and mental health service staff across the region. Additional considerations identified in this process included the need to accentuate the library as additional to (rather than alternative to) PSW roles, and the need to increase diversity and access to a broad range of living books across age, ethnicity, gender, sexuality and types of experience.

The research team includes two PPI leads with lived experience (CL and KM). They contributed to the design of the study and application for funding. They are involved in strategic management meetings and practical delivery in both workstreams, including cofacilitating the EBCD workshops. They will be authors on all outputs and involved in dissemination including presentations, and as coapplicants for future grants.

Our expert group includes two people with lived expertise. One is a service user who has been a reader (or book) in a living library, and the other is a carer who has a lot of experience in service development at an advisory level.

## Knowledge mobilisation

Outputs from this study will include an expert informed theory of how a LoLEM could work to improve outcomes for books and readers, and a theory-informed codesigned protocol for setting up and running a LoLEM. These will be openly shared in open access journals, following RAMESES reporting guidelines[52] and through Lancaster University and Lancashire and South Cumbria NHS websites. Any amendments made to the protocol will be explained in the outcome paper on completion of the study, and updated on PROSPERO.

We also aim to have established a network of people with lived experience and mental health support providers who are motivated to use the study outputs to establish a library. We will host a dissemination event to present the findings of the study to stakeholders including commissioners. We will invite mental health support providers, commissioners, health professional trainers and other organisations who may benefit from hosting a LoLEM. We hope to use this to build a collaborative twice daily for further research funding to conduct a realist evaluation of a series of LoLEMs across different contexts.

## ETHICS AND DISSEMINATION

In the same way that a poorly designed LoLEM could be disempowering, stressful and even traumatising for books, and could confirm rather than challenge existing stigma and prejudice, this could also be true for the process of codesigning the library guide. To mitigate against this, we will conduct the study within a university (rather than NHS) context; ensure all EBCD workshops will be cofacilitated with people who have lived expertise; and ensure all participants will have access to one to one support from the research team throughout the process as required. EBCD group members will be purposively sampled to ensure a diverse range of views and minimise any power issues. We will encourage exploration of the role of a LoLEM within the broader social context of mental health support, as well as more detailed design features of the library.

All data will be provided with informed written consent and will be collected and stored securely. The detailed and personal nature of the EBCD workshop data makes it impossible to fully anonymise, so these data will not be shared openly. However, all outputs will be published in open access journals. Plain English summaries will be made available on the Lancaster University website.

## Author affiliations
[1]Division of Health Research, Lancaster University, Lancaster, UK
[2]Lancaster University Library, Lancaster University, Lancaster, UK
[3]Freelance participatory artist, Lancaster, UK
[4]Health Education England North West, Manchester, UK
[5]Patient Experience, Engagement & Safeguarding, Lancashire and South Cumbria NHS Foundation Trust, Preston, UK
[6]Independent survivor researcher, Lancaster, UK
[7]Department of Psychiatry, The University of British Columbia, Vancouver, British Columbia, Canada
[8]Lancaster Medical School, Lancaster University, Lancaster, UK
[9]Faculty of Health and Medicine, Lancaster University, Lancaster, UK
[10]Institue of Mental Health, University of Nottingham School of Health Sciences, Nottingham, UK
[11]Faculty of Medicine and Health Sciences, Nord University, Namsos, Norway

**Acknowledgements** MS acknowledges the support of the NIHR Nottingham Biomedical Research Centre.

**Contributors** FL is the guarantor; led the design and funding application, and co-led the running of the project. PM is an early career researcher who led the ethics application and the realist synthesis and integration of theory between the workstreams; and co-led the management of the study. JB advised on the search strategy. GC cofacilitated EBCD workshops and produced artistic impressions of the developing theory. ZG, PJ, RJ and LW will help with the running of the EBCD workshops, and RJ also screened papers and extracted data in the realist synthesis. CL and KM will cofacilitate the EBCD workshops, with KM leading the codesign process and CI supporting public and patient involvement. JR-M provided methodological guidance and input into the realist synthesis. SHJ was organisational lead for the EBCD workstream and was consulted on theory development in the realist synthesis. SF, CI, HK, SP, EM and MS provided strategic guidance to the design of the study. All authors contributed to and approved the final version of all protocols.

**Funding** The study was funded by the National Institute of Health and Care Research (NIHR), Research for Patient Benefit programme (NIHR203476). The views expressed are those of the authors and not necessarily those of the NIHR or the Department of Health and Social Care. The study is sponsored by Lancashire and South Cumbria NHS Foundation Trust.

**Competing interests** None declared.

**Patient and public involvement** Patients and/or the public were involved in the design, conduct, reporting or dissemination plans of this research. Refer to the Methods and analysis section for further details.

**Patient consent for publication** Not applicable.

**Ethics approval** This study involves human participants and was approved by Ethical approval was granted by Coventry and Warwick National Health Service (NHS) Research Ethics Committee (REC) on 29th December 2021 (REF: 305975). Participants gave informed consent to participate in the study before taking part.

**Provenance and peer review** Not commissioned; externally peer reviewed.

**Data availability statement** No data are available. n/a.

**ORCID iDs**
Zoe Glossop http://orcid.org/0000-0002-6151-4860
Sarah Powell http://orcid.org/0000-0002-3509-5789

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
