## [Reviewer comments · BMJ Open]

ARTICLE DETAILS

TITLE (PROVISIONAL)	Designing a Library of Lived Experience for Mental Health (LoLEM): Protocol for integrating a realist synthesis and Experience Based Co-Design approach
AUTHORS	Lobban, Fiona; Marshall, Paul; Barbrook, John; Collins, Grace; Foster, Sheena; Glossop, Zoe; Inkster, Clare; Jebb, Paul; Johnston, Rose; Khan, Hameed; Lodge, Christopher; Machin, Karen; Michalak, Erin; Powell, Sarah; Rycroft-Malone, Jo; Slade, Mike; Whittaker, Lesley; Jones, Steven

VERSION 1 – REVIEW

REVIEWER	Wong, Geoff Oxford University, Nuffield Department of Primary Care Health Sciences
REVIEW RETURNED	09-Oct-2022

GENERAL COMMENTS	I read this manuscript as a realist review methodologist and not as someone who has any content expertise or expertise of experience based co-design. I have predominantly focused my peer review on the methodological aspects of the realist review. Overall, I found this protocol is clearly written and sets out what is planned in logical way. The use of a realist review approach and addition of experience based co-design is appropriate and clearly justified. I have a few minor comments for the authors to address or consider: --Aims and objectives - page 9 of 22 of pdf. Would be informative to readers if you provided your research question(s). --On page 9 of 22 lines 50 to 51 you mention that your study is underway. Please would you clarify how much of the review and EBCD work has already taken place. My concern here is whether, at this stage, you are in a position to change anything based feedback from peer-reviewers. --In section 4. Identifying relevant literature (pages 11 to 13 of 22 of pdf), you may want to consider the option of doing additional searches where needed. This may help you to 'fill in the gaps' in your programme theory - given that your initial scoping searches have not retrieved that many relevant documents. Off the top of my head (for example), it may emerge that authenticity of the living library in the eyes of a person with mental health is an important issue. This is something which is also likely to be relevant to peer supporters and hence something may be learnt from looking in that literature. However, I accept that you will be collecting some primary data and hence the need to use additional searches to plug 'gaps' in your programme theory may not be as
---

	important. --Section 7 page 14 of 22 of pdf. To avoid confusion and for sake of consistency, it may be worth changing "middle range theory" in this instance to "formal" or "substantive" theory. See: https://www.ramesesproject.org/media/RAMESES_II_Theory_in_realist_evaluation.pdf --Section 8 page 14 of 22 of pdf. It would be of interest to readers to know when you plan to do the Stakeholder interviews (i.e how far into your review) and also with how many people? If you are going to use the interviews as one of your strategies to plug 'gaps' in your programme theory, you will only know what these 'gaps' are once you have progressed your realist review. Also, knowing what the 'gaps' are will also inform who you wish to sample - as different people will know different things about living libraries. --Good luck with your interesting project.
--	---

REVIEWER	Janes, Ed Cardiff University, CASCADE, SOCSI
REVIEW RETURNED	16-Nov-2022

GENERAL COMMENTS	Many thanks for inviting me to review this interesting and ambitious protocol. The objectives are clear and (as someone who is not a expert on the topic) the background literature outlines the topic, advances and impacts of living libraries. The design being used in the study is appropriate, but I feel that the protocol itself needs some work. There are two main issues that interlink. 1)How the methods link is a little confusing – Yes, you are concurrently developing a realist synthesis/theory, and using EBCD – but I think this needs a refinement stage. Combining the EBCD within the realist synthesis will not work 2)Even before including the EBCD component, realist studies have lots of steps – your transitions through the process need to be tighter. As a result, I have highlighted some minor amendments for consideration – the attached documents includes some more detailed comments with line references.
--

REVIEWER	Jenkin, Gabrielle University of Otago Wellington, Deans Department
REVIEW RETURNED	13-Dec-2022

GENERAL COMMENTS	Thank you for the opportunity to review this important study protocol. It is well written with a good rationale. However, the following points need clarification: Can you define a person with lived experience - surely most of us have lived experience of mental distress/illness in some capacity/ as carer or as affected personally? Or are you meaning those with diagnosed mental illness that choose to identify (and brave enough) as having lived experience? It is not clear until 3/4 way through the protocol that people with lived experience are on the research team. Who is on the advisory group - is this lived experience people as well? People with lived experience should ideally be involved all the way through - even on data interpretation if possible. It is not clear throughout the protocol.
---

	In terms of the literature search - I recommend you include grey literature as there is quite a bit in the mental health field that includes the often-missing voices of people with lived experience - that may not be published or included in other systematic reviews due to publication bias and other things. Also I think the review should include reports/publications regardless of whether there was a formal evaluation for the same reason - and some will list learnings although may not have been formally evaluated. So maybe make your inclusion and exclusion criteria clearer? In terms of the realist evaluation method - there is not much definition of the WHO - what different types of people are you going to consider living libraries are good for, by gender, ethnicity socio-economic status, the general public or clinical populations, those in the community or those in institutions, those with serious or moderate mental illnesses? These are as important as context. Sensitivity of the topic matter will also be an important/maybe a critical contextual factor - e.g people with lived experience of suicide attempt, people with lived experience of HIV, lived experience of racial or other discrimination or bullying etc. And these will vary by culture, how are you planning on dealing with these? Minor point is that there are a few missing words - so check the manuscript for these before publishing. I very much commend the authors for this great study protocol and idea however.
--	---

VERSION 1 – AUTHOR RESPONSE

Reviewer: 1

Dr. Geoff Wong, Oxford University

Comments to the Author:

I read this manuscript as a realist review methodologist and not as someone who has any content expertise or expertise of experience based co-design.

I have predominantly focused my peer review on the methodological aspects of the realist review.

Overall, I found this protocol is clearly written and sets out what is planned in logical way.

The use of a realist review approach and addition of experience based co-design is appropriate and clearly justified.

I have a few minor comments for the authors to address or consider:

--Aims and objectives - page 9 of 22 of pdf.

Would be informative to readers if you provided your research question(s).

Added as requested

“Our Research Questions are:

1. What are the impacts of a LoLEM, on books and readers? How do these impacts happen, for whom, and in what contexts?

2. What are the key aspects to consider in designing a LoLEM? “

--On page 9 of 22 lines 50 to 51 you mention that your study is underway.

Please would you clarify how much of the review and EBCD work has already taken place.

My concern here is whether, at this stage, you are in a position to change anything based feedback from peer-reviewers.

The study is underway but not yet complete. In the review, we have identified relevant literature and completed the data extraction. At the point of submitting the paper we were in step 4 (identifying relevant literature). We are currently in the phase of data analysis (step 6), and we are very happy to adapt this process and steps 7-9 in response to feedback. In the EBCD, we have completed the workshops and we are working on a draft of the implementation plan, which we will continue to adapt in light of feedback, even after the end of the study. We will offer a draft of the guide for piloting and consider this a working document which will be continuously revised in light of feedback.

--In section 4. Identifying relevant literature (pages 11 to 13 of 22 of pdf), you may want to consider the option of doing additional searches where needed.

This may help you to 'fill in the gaps' in your programme theory - given that your initial scoping searches have not retrieved that many relevant documents.

Off the top of my head (for example), it may emerge that authenticity of the living library in the eyes of a person with mental health is an important issue. This is something which is also likely to be relevant to peer supporters and hence something may be learnt from looking in that literature.

However, I accept that you will be collecting some primary data and hence the need to use additional searches to plug 'gaps' in your programme theory may not be as important.

Thank you for this suggestion. Consistent with this, in step 7 (Draw on middle range theories) we have included the statement

“At this stage, and consistent with the iterative nature of realist synthesis methodology, we may conduct additional theory-driven literature searches for further middle range theories that augment the theory development process. The selection and use of middle range theories will be guided by the extent to which each theory helps us to further develop our programme theories.”

This allows us to conduct additional searches as you suggest. In fact, we have already identified the concept of “psychological safety” as particularly relevant to our evolving theories and we are considering running an iterative search to examine this further. We will report any additional searches that emerge from the iterative process in the final paper reporting on the results of the protocol.

--Section 7 page 14 of 22 of pdf.

To avoid confusion and for sake of consistency, it may be worth changing "middle range theory" in this instance to "formal" or "substantive" theory.

See:

https://eur02.safelinks.protection.outlook.com/?url=https%3A%2F%2Fwww.ramesesproject.org%2Fmedia%2FRAMESSES_II_Theory_in_realist_evaluation.pdf&data=05%7C01%7Cf.lobban%40lancaster.ac.uk%7C9a7e466ccc2040bcc30708daddf4ef1d%7C9c9bcd11977a4e9ca9a0bc734090164a%7C0%7C0%7C638066341286955972%7CUnknown%7CTWFpbGZsb3d8eyJWIjoiMC4wLjAwMDAiLCJQIjoiV2luMzliLCJBTiI6IklhaWwiLCJXVCi6Mn0%3D%7C3000%7C%7C%7C&sdata=3V4G47ZdnY7RUqHl3jyo%2F1rTY2xRxaM9DfpmRpOa%2Bk%3D&reserved=0

Thank you for this helpful clarification – references to middle range theories have been replaced by formal theories.

--Section 8 page 14 of 22 of pdf.

It would be of interest to readers to know when you plan to do the Stakeholder interviews (i.e how far into your review) and also with how many people?

If you are going to use the interviews as one of your strategies to plug 'gaps' in your programme theory, you will only know what these 'gaps' are once you have progressed your realist review.

Also, knowing what the 'gaps' are will also inform who you wish to sample - as different people will know different things about living libraries.

To address this point and for further clarification, we have added the following to page 13

“Early interviews will inform the development of our initial programme theories, but as our theories develop, the focus of the interviews will evolve into a more specific focus on filling in gaps in our theories, and testing competing theories. Sampling will become more purposive to select interviewees with the relevant knowledge and expertise.”

--Good luck with your interesting project.- **Thank you**

Reviewer: 2

Dr. Ed Janes, Cardiff University

Comments to the Author:

Many thanks for inviting me to review this interesting and ambitious protocol. The objectives are clear and (as someone who is not an expert on the topic) the background literature outlines the topic, advances and impacts of living libraries.

Thank you

The design being used in the study is appropriate, but I feel that the protocol itself needs some work. There are two main issues that interlink. 1)How the methods link is a little confusing – Yes, you are concurrently developing a realist synthesis/theory, and using EBCD – but I think this needs a refinement stage. Combining the EBCD within the realist synthesis will not work 2)Even before including the EBCD component, realist studies have lots of steps – your transitions through the process need to be tighter. As a result, I have highlighted some minor amendments for consideration – the attached documents includes some more detailed comments with line references.

From attached doc

Page / Line	Comment
P8; 38-53	As a short summary of realist approaches this is mostly fine but is worth explaining that there are multiple mechanisms in a system, each with context and outcomes. Also I wouldn't equate mechanisms to 'programme resources' – mechanisms are broader than that.
	We agree with both points. We have added an s to ways – to make it clear that there are multiple mechanisms in a system, as well as multiple contexts. The mechanisms in the sentence below refers to the “ways in which people respond to the programme resources offered (mechanisms-M)” rather than to the programme resources per se. “The approach is grounded in realist philosophy and proposes that complex interventions in health have intended and unintended impacts (outcomes-O) through the ways in which people respond to the programme resources offered (mechanisms-M). The extent to which these mechanisms are triggered depends on specific features of the environment in which the programme is delivered, or characteristics of the participants (contexts-C).”
P11; 17 – P12; 44	Be clear that this is all the scoping review – keep as 1.
	We have made the distinction between workstream 1 – the scoping review, and workstream 2 – the EBCD workshops, more distinct by using higher level headings to more clearly differentiate the workstream headings from the subheadings within each workstream.
P11; 20-25	Might be useful to reconsider the language used here – 'interviews' suggest data collection and this would no longer be a review of past studies. Maybe

	is more about experiences informing statements?
	The realist synthesis methodology explicitly recommends the use of interview data to inform the development of initial programme theories, and their refinement in a realist synthesis. Following this methodology, we have not made a change here as we did intentionally use the term interviews “We will draw on lived experiences of the research team and Expert Group to develop our initial programme theories. We will first conduct individual interviews with members who have hosted a living library (CI), or been a human book (CL, HK), or a reader (FL, SJ), and refine emerging theories in group discussion.”
P12; 34-42	You need either inclusion AND exclusion criterion for all (eg IC = Full text available in English vs EC = Not English; Partial text) or just the inclusion for all – reverse the last to be inclusive
	Done – we have modified Table 2 accordingly, replacing an exclusion criterion for non-written files with an inclusion criterion for “written text only”
P12; 54-55	The detail on searches, screening etc is good overall but this needs clarity – if the study is high quality and theory strong would that give confidence in the result?
	Yes – we have added the following sentence to clarify this “Only studies considered to be relevant and of sufficient rigour for us to trust the implications for our theory development will be included”
P13; 29	Once you integrate the workstreams, this is no longer based on literature alone – this is theory refinement. Separate into new section
	In realist synthesis methodology, theory development and refinement are not distinct stages, and both can be informed by the collection of primary data including interviews, and, in this case, EBCD workshop data. However, we have made attempts to delineate the two workstreams more clearly, using clearer headings to achieve this (see change details above).
P16; 9	Mention RAMESES when introducing the realist synthesis and the method used.
	The RAMESES guidelines on realist synthesis are publication standards which will be followed and guide the reporting of the findings. This is already stated and referenced under the section on Knowledge Mobilisation “Outputs from this study will include an expert informed theory of how a LoLEM could work to improve outcomes for books, and readers; and a theory informed co-designed protocol for setting up and running a LoLEM. These will be openly shared in open access journals, following

	RAMESES reporting guidelines (52) and through Lancaster University and Lancashire and South Cumbria NHS websites” Wong G, Greenhalgh T, Westhorp G, Buckingham J, Pawson R. RAMESES publication standards: realist syntheses. BMC medicine. 2013;11(1):21
	Overall needs a final proofread/sensecheck
	This has been completed and minor edits corrected

Best Wishes
Ed

Reviewer: 3

Dr. Gabrielle Jenkin, University of Otago Wellington Comments to the Author:

Thank you for the opportunity to review this important study protocol.

It is well written with a good rationale.

Thank you

However, the following points need clarification:

Can you define a person with lived experience - surely most of us have lived experience of mental distress/illness in some capacity/ as carer or as affected personally? Or are you meaning those with diagnosed mental illness that choose to identify (and brave enough) as having lived experience?

It is very challenging to create precise definitions around mental health or what kind of lived experience other people might usefully learn from. We have deliberately not defined this by diagnosis. Some people do not receive a diagnosis, despite very challenging difficulties. Others may avoid a diagnosis by choosing not to engage with mental health services. Either of these may be interesting books for readers facing similar challenges. We agree that most of us have of mental distress/illness in some capacity/ as carer or as affected personally, and there could be many benefits from more of us openly sharing these experiences. However, our experience of working in the peer support field is that not all of us have a story we feel compelled to share, or feel drawn to use our experience to help others. The aim of the LoLEM guide designed in workstream 2 is that it highlights the key decisions that need to be made when setting up a LoLEM across a range of different contexts. One of the key decisions will be – who are the books? And who are the readers? The precise definition of what kind of lived experience is likely to be helpful, will depend on the aims of the library and the targeted readers.

We have added clarification on this to page 15

“Key issues anticipated include: what the library is for, where and how the library is hosted, how books are defined, recruited, trained, reimbursed, supported; who the readers are, and how the library is offered to them; how the role of the librarian is set up, and how a LoLEM should be evaluated. “

We have also added a clearer definition of criteria for participants taking part in the EBCD workshops (pg 14)

“Participants will include people with experience of using their mental health experiences to support others or to inform research and development in health services; and mental health professionals with expertise in supporting peer workers.”

It is not clear until 3/4 way through the protocol that people with lived experience are on the research team. Who is on the advisory group - is this lived experience people as well? People with lived experience should ideally be involved all the way through - even on data interpretation if possible. It is not clear throughout the protocol.

The research team and expert group are described in detail on page 9. Additional details have been added to clarify that there are people with lived experience embedded within both of these.

“Our team includes people with experience in group facilitation and peer support for mental health (KM, CL), mental health professionals (PJ, LW), clinical academics (FL, SJ, JRM), information specialists (JB), researchers (PM, RJ, ZG) and an artist (GC). We have established a non-independent Expert Group to oversee and guide the project. This includes five people with expertise in hosting (EEM, CI, SP), researching (MS), or taking part in a living library as a book with experience in managing mental health challenges (HK), and a mental health carer with a broad interest in lived expertise frameworks (SF). The group will be consulted at each of the steps described below throughout the study, via monthly online video meetings.”

Further detail is given within the PPI section which we have moved to the end of the methods section as requested by the editor.

In terms of the literature search - I recommend you include grey literature as there is quite a bit in the mental health field that includes the often-missing voices of people with lived experience - that may not be published or included in other systematic reviews due to publication bias and other things.

We agree and have included grey literature. This is stated on page 10

“Grey literature will be searched using: ProQuest (theses/conference documents), British Library via Ethos (theses), International Clinical Trials Registry Platform (clinical trials), Overton.io (UK health policy), Bielefeld Academic Search Engine (articles/these/conference documents), Google.com*

We will conduct google searches of the following domains for websites that included the words “human library” or “living library”. - Inurl: ac (academic institutions) - Inurl: edu (academic institutions) - Inurl: gov (government-published) - Inurl: org (non-governmental organisations) - Inurl: nhs (UK National Health Service). “

Also I think the review should include reports/publications regardless of whether there was a formal evaluation for the same reason - and some will list learnings although may not have been formally evaluated. So maybe make your inclusion and exclusion criteria clearer?

We agree. We have included all reports / publication types, and there is no requirement for formal evaluation. This is outlined in page 11

“We will include a wide range of data sources including primary data, reviews, commentaries, manuals, and detailed lived experience accounts. Inclusion and exclusion criteria are listed in table 2.

Table 2: Inclusion and exclusion criteria for living library systematic search.

Inclusion criteria
- Full-text available in English
- Published since 2000 (inception of the human library concept)

- Relates to human or living library (as per the above definition)
- Any study type is permitted
Exclusion criteria
- Non-written files (audio-visual content)

“

In terms of the realist evaluation method - there is not much definition of the WHO - what different types of people are you going to consider living libraries are good for, by gender, ethnicity socio-economic status, the general public or clinical populations, those in the community or those in institutions, those with serious or moderate mental illnesses? These are as important as context.

Again, we agree that understanding who might benefit from a living library model is crucial. At this stage we don't know the answer, but it is a key focus of the study. Therefore, we are using a realist synthesis to first understand if the existing literature and stakeholder interviews can tell us anything about who a living library model is likely to work for. Hence our first research question is stated as:

1. What are the impacts of a LoLEM, on books, readers and librarians? How do these impacts happen, for whom, and in what contexts?

Based on our scoping, we anticipate that the living library model literature is unlikely to tell us definitely how the outcomes will differ across gender, ethnicity, socioeconomic status, or different clinical populations. However, where this is evidenced, this will inform our theories.

In workstream 2, the EBCD process will allow us to explore who the codesign group think might benefit from a living library and therefore where they should be offered.

As stated on page 15, this is likely to be a key issue addressed in the LoLEM guide.

“The aim is not to write a “recipe book”, but to identify key things to consider when setting up a library, and to understand how different decisions are likely to lead to different outcomes. These will be described using text and artistic impressions. Key issues anticipated include: what the library is for, where and how the library is hosted, how books are defined, recruited, trained, reimbursed, supported; who the readers are, and how the library is offered to them; how the role of the librarian is set up, and how a LoLEM should be evaluated.”

Following this project, we would like to test use of the LoLEM guide across a wide range of contexts using a realist evaluation approach. This design would allow us to test who the library works best for.

Sensitivity of the topic matter will also be an important/maybe a critical contextual factor - e.g people with lived experience of suicide attempt, people with lived experience of HIV, lived experience of racial or other discrimination or bullying etc. And these will vary by culture, how are you planning on dealing with these?

We could not agree more. The primary purpose of the living library model is to allow complex sensitive issues to be discussed, and to explore different perspectives of people from a range of cultures and backgrounds. The way in which these issues will be managed, will be addressed in the LoLEM guide – which is the output of the codesign process in this study. Therefore at the protocol stage, we cannot define what this will be. These issues will be addressed in detail in the guide, so that anyone using the LoLEM guide to run a library in future studies will have the opportunity to consider all of these issues in detail during the design phase.

Minor point is that there are a few missing words - so check the manuscript for these before publishing.

Proof read again and minor edits made

I very much commend the authors for this great study protocol and idea however.

Thank you

Reviewer: 1

Competing interests of Reviewer: I have no competing interests to declare.

Reviewer: 2

Competing interests of Reviewer: None

Reviewer: 3

Competing interests of Reviewer: None

VERSION 2 – REVIEW

REVIEWER	Wong, Geoff Oxford University, Nuffield Department of Primary Care Health Sciences
REVIEW RETURNED	22-Jan-2023

GENERAL COMMENTS	Thank you for taking the time to satisfactorily address my comments. I have no further comments.
--

REVIEWER	Janes, Ed Cardiff University, CASCADE, SOCSI
REVIEW RETURNED	09-Feb-2023

GENERAL COMMENTS	Apologies for the slightly late review. Many thanks for sending me the revised manuscript, and for making the revisions so easy to see and assess. In terms of the minor suggestions given in the first review, I'm happy that these have either been made, or that the authors have considered them and made a case for keeping the manuscript as it was. In particular, improvements have been made to the overall structure - it is clear from a final read through of the whole manuscript that the different stages of the complex procedure being better delineated. Overall, I'm pleased to recommend this be accepted for publication.
---